# Single-Walled Carbon Nanohorns as Boosting Surface for the Analysis of Low-Molecular-Weight Compounds by SALDI-MS

**DOI:** 10.3390/ijms23095027

**Published:** 2022-04-30

**Authors:** Marco Roverso, Roberta Seraglia, Raghav Dogra, Denis Badocco, Silvia Pettenuzzo, Luca Cappellin, Paolo Pastore, Sara Bogialli

**Affiliations:** 1 Department of Chemical Sciences, University of Padova, IT35131 Padova, Italy; raghav.dogra@studenti.unipd.it (R.D.); denis.badocco@unipd.it (D.B.); silvia.pettenuzzo-1@unitn.it (S.P.); luca.cappellin@unipd.it (L.C.); paolo.pastore@unipd.it (P.P.); sara.bogialli@unipd.it (S.B.); 2 Institute of Condensed Matter Chemistry and Technologies for Energy (ICMATE), National Research Council-CNR, IT35127 Padova, Italy; roberta.seraglia@cnr.it; 3 Center Agriculture Food Environment (C3A), University of Trento, IT38098 San Michele all’Adige, Italy

**Keywords:** nanohorns, SALDI, MALDI, mass spectrometry, amino acids, triglycerides

## Abstract

Limits of Matrix-Assisted Laser Desorption Ionization (MALDI) mass spectrometry (MS) in the study of small molecules are due to matrix-related interfering species in the low *m/z* range. Single-walled carbon nanohorns (SWCNH) were here evaluated as a specific surface for the rapid analysis of amino acids and lipids by Surface-Assisted Laser Desorption Ionization (SALDI). The method was optimized for detecting twenty amino acids, mainly present as cationized species, with the [M+K]^+^ response generally 2-time larger than the [M+Na]^+^ one. The [M+Na]^+^/[M+K]^+^ signals ratio was tentatively correlated with the molecular weight, dipole moment and binding affinity, to describe the amino acids’ coordination ability. The SWCNH-based surface was also tested for analyzing triglycerides in olive oil samples, showing promising results in determining the percentage composition of fatty acids without any sample treatment. Results indicated that SWCNH is a promising substrate for the SALDI-MS analysis of low molecular weight compounds with different polarities, enlarging the analytical platforms for MALDI applications.

## 1. Introduction

The analysis of low molecular weight compounds by matrix-assisted laser desorption ionization mass spectrometry (MALDI-MS) is challenging when common organic MALDI matrices and traditional sample deposition procedures are used. The main issue is related to the presence of background interfering species, generated by the ionization of the matrix itself, which are accounted for the suppression of analytes signals in the low mass region of the spectrum. Furthermore, the co-crystallization process, i.e., the inclusion of analyte molecules in the matrix crystals occurring by solvent evaporation at room temperature, causes inhomogeneity of the resulted crystals, the presence of hot spots, and the unwanted coffee-ring effect [1]. The severity of these effects is matrix-dependent and interferes with the possibility of obtaining accurate quantitative results and good reproducibility by MALDI-MS analysis.

In recent decades, many efforts were performed to achieve the production of ions from low molecular weight analytes without the presence/assistance of matrices and avoiding/reducing the severe decomposition reactions typical of laser desorption ionization (LDI) conditions. In this frame, interesting results were obtained by the deposition of analytes on specific surfaces, chemically and physically different from the stainless steel-based usually employed as support for the samples, giving rise to a new analytical technique known as surface-assisted laser desorption ionization mass spectrometry (SALDI-MS).

The nanomaterial-based surface showed great potential in replacing common organic matrices for the ionization of molecules in SALDI conditions [2,3]. Various applications are reported in the literature, generally based on nanoparticles (NPs) with different chemical characteristics, and based on materials such as Au, Pt, iron oxide, silica, titanium oxide and carbon. The ionization mechanism implied in the production of analyte ions is still debated and not fully understood. Generally, it is reported that the laser shot promotes the formation of a laser-induced plasma containing ions or activated chemical species that in turn interact with the sample, provoking the ionization of analytes. The surface plasmon effect and thermal and non-thermal processes may also be involved in the ionization process, contributing to the desorption of non-volatile analytes and to the final efficiency of the technique [4]. Performances of different materials in SALDI applications in terms of sensitivity, reproducibility and mass range of applicability, depend on several parameters, i.e., size, morphology, superficial area, molar absorption coefficient, heat capacity, thermal and electrical conductivity, possible functionalization and affinity with the analyte [5]. Furthermore, SALDI surfaces generally showed good salt tolerance [6] and negligible fragmentation of analytes [5].

Carbon-based NPs are promising materials for SALDI applications [7]. Despite general requirements, carbon materials are cost-effectiveness, functionalizable and suitable, in terms of affinity, to be used for the analysis of various biomolecules and environmental pollutants [6]. Graphite, fullerenes, nanotubes and diamond-based surfaces were successfully employed in the analysis of proteins, oligonucleotides, synthetic polymers, fatty acids, drugs, pesticides and other environmentally relevant compounds, as reviewed by Najam-ul-Haq et al. [8]. Notably, carbon nanowalls (CNWs) treated by atmospheric pressure plasma to increase hydrophobicity were investigated for amino acids analysis [9], while Sakai et al. demonstrated that the signal to noise ratio of arginine was increased, and the fragmentation of analyte reduced, using CNWs with a narrower wall-to-wall distance, testing distances from 142 to 467 nm [5].

Graphene oxide (GO) and GO derivatives were applied to the analysis of different environmental pollutants, peptides, amino acids, fatty acids, nucleosides and nucleotides and flavonoids. Moreover, GO/carbon nanotubes and GO/Au hybrid structures were reported for analyzing amino acids and saccharides [6]. Recently, other carbon-based nanomaterials were proved to be very efficient for SALDI-MS applications. Carbon NPs obtained from candle soot were applied for the analysis of polymers (polyethyleneimine and poly(ethylene glycol), pharmaceuticals (verapamil, methadone), peptides (glutathione, bradykinin 1–7 and bacitracin) and glucose present at very low concentration in complex biological matrices [10]. Carbon dots (CDs) and N-doped CDs were positively evaluated for the analysis of cholesterol [11], while CDs doped with nitrogen and sulfur were developed and tested for the determination of the distribution of bisphenol S in mouse tissues by imaging-MS [12]. Propranolol was quantitatively determined in mussels by SALDI-MS adopting an oxidized carbon black-based surface [13], while mesoporous graphene was used as an enrichment material and matrix for SALDI analysis of polyphenols in rat plasma and urine samples [14]. Graphite carbon black was successfully used for the direct and simultaneous absorption and SALDI-MS analysis of taste- and odor-active compounds (ethyl esters, aldehydes, alcohols, fatty acids, lactones, amino acids and sugars) in liquid samples [15]. Composite materials, made of carbon nanotubes and polymeric nanofibers were used as a substrate for the SALDI analysis of small drug molecules and high molecular weight synthetic polymers and proteins reaching detection limits ten times lower than traditional MALDI-MS [16]. In addition, a substrate composed of polyamide and carbon soot nanoparticles was applied to the determination of methadone in urine and residual malachite green on fish skin showing high sensitivity, good reproducibility and a limit of detection lower than 200 pmol [17]. Boronic acid-functionalized magnetic multi-walled carbon nanotubes were also proposed for the analysis of flavonoids in food matrices, obtaining nmol/L detection limits and good recoveries [18].

Single-walled carbon nanohorns (SWCNHs) are horn-shaped single-walled tubules with a conical tip, self-organized in ‘dahlia-like’ aggregates. The synthesis is usually performed by laser ablation of pure graphite without using metal catalysts [19]. SWCHNs show a wide range of potential applications, i.e., gas storage, adsorption, catalyst support, drug delivery system, biosensing, fuel cells and many others [19,20]. SWCNH aggregates exhibit 80–100 nm diameter, high surface area and good thermal and electrical conductivity, making them optimal potential substrates for SALDI-MS. In this light, bare SWCNHs were preliminary tested for the analysis of peptides and fatty acids, and evaluated for the quantitative analysis of antineoplastic drugs (irinotecan, sunitinib, and 6-a-hydroxy-paclitaxel) for pharmacokinetic studies in human blood plasma and personalized medicine applications [21], while aptamer functionalized SWCNHs were developed for the analysis of ATP [22].

The present work reports the optimization of a SALDI-based method for the analysis of low molecular compounds with a wide range of polarity, in particular, several amino acids and triglycerides. The single-walled carbon nanohorns-based surface was tested under different conditions to describe from an experimental point of view the ionization processes occurring on the substrate and the combined effect of analyte and surface on the production of ionic adducts during the process. The method was also applied to the analysis of real samples i.e., olive oil, to propose possible future applications.

## 2. Results and Discussion

### 2.1. Evaluation of Characteristics and Performances of the MS-Surface

The reported surface preparation method required simple steps and can be completed within a few minutes. The surface is ready to be used and is stable for at least seven days without taking precautions. Prepared surfaces were not washable for reuse, in order to avoid loss of material, and for this reason, each analyzed sample required the preparation of its own surface. LDI spectra of SWCNH-based surfaces were acquired at different laser intensity, from 5% to 50%, in both positive and negative ionization modes, to depict and evaluate the presence of potentially interfering signals in the low *m*/*z* range, generally due to the ionization of residual contaminants on the sample-holder or contained in solvents and chemicals used. The ionization of these species, also possible in MALDI conditions, may be responsible for analytes ion suppression or signal overlapping. A comparison of the spectra obtained at different laser power is reported in Figure 1. At quite low values of laser intensity, namely 5–20%, interfering species are negligible, and only low-intensity signals up to *m*/*z* 400 are detectable in both positive and negative ionization mode, probably due to residuals present on the stainless-steel sample holder (Figure 1, panels A). Increasing the laser power to 25%, the LDI spectrum (Figure 1, panel B) showed an additional series of low abundant signals in the 700–4500 *m*/*z* range, which intensities greatly increased by setting the laser power to 50% (Figure 1, panel C). Over the value of about *m*/*z* 800, the detected signals were separated by 24.001 mass units, ascribable to two carbon atoms, putatively due to the decomposition of SWCNHs as a result of the interaction with the laser beam. At a lower mass range in positive ionization mode, it was also possible to detect signals ascribable to sodium and potassium cations. These signals were still detectable even after repetitive washing of the surface with ultrapure water, meaning that sodium and potassium impurities are strictly embedded inside the SWCNHs structure and are difficult to be removed. Considering the obtained data, differently from the common MALDI matrices, this surface is potentially applicable to the analysis of low molecular species maintaining the laser power lower than 50%, thus avoiding strong interfering signals due to the ionization of SWCNHs itself. It is also to highlight that ion source performances decrease faster compared to the use of common MALDI matrices, and for this reason, the ion source required a more frequent routine cleaning. Another drawback is related to the fact that the SWCNH-based surface is black and is not easy to distinguish the pic produced by the camera of the instrument from the sample holder.

### 2.2. Analysis of Amino Acids

The analysis of free amino acids is challenging in MALDI conditions, as the ionization of the matrix is involved in the suppression of analyte signals in the low *m*/*z* range. In this light, the SWCNH-based surface was tested to evaluate the ionization effectiveness in the analysis of the 20 natural amino acids.

As an example, Figure 2 reports the spectra, in positive ionization mode, obtained in the case of glycine (panel A) and phenylalanine (panel B). Both analytes were detected as cationized adducts with sodium ([M+Na]^+^) and potassium ([M+K]^+^), with the potassium adduct generally significantly higher in intensity than the sodium one. The expected protonated ion species [M+H]^+^ was barely detectable only for phenylalanine with very low intensity at *m*/*z* 166.090. Other ionic species with low intensities reported in the spectra are due to [M−H+2Na]^+^, [M−H+2K]^+^ and [M−H+Na+K]^+^ adducts. As a comparison, Figure 2 panel C shows that it was not possible to detect ionic signals ascribable to glycine and phenylalanine in MALDI condition using the conventional α-Cyano-4-hydroxycinnamic acid (CHCA) matrix, which exhibits a number of interfering signals. A detailed identification of the species present in the spectra in Figure 2 is reported in Appendix A. This behavior, i.e., the [M+K]^+^ > [M+Na]^+^ >> [M+H]^+^, was also observed for the other analyzed amino acids where the protonated species is negligible or present with very low intensity (Figure 3). The only exception is arginine, which protonated ion was larger than the sodium adduct, though the potassium adduct was the most intense signal. This finding can be explained by the higher relative basicity of arginine due to its guanidinic moiety.

Considering the evidence of the residual amount of Na and K on the SWCNHs surface previously described for the interfering species, and in order to explain the higher abundance of the potassium adduct compared to the sodium one, SWCNH were analyzed by ICP-MS to quantify these metals. The concentration of sodium was 0.27 mg/g (0.012 mmol/g), while potassium was present at 1.48 mg/g (0.028 mmol/g). SWCNHs contain potassium at about double the concentration of sodium, and therefore, the formation of the potassium adduct is favored. Anyway, since the percentage of the adducts was not equal for all amino acids, a deepened evaluation of the relative ionization efficiency with sodium was attempted considering some chemo-physical properties of the selected amino acids that may have a relationship with the formation of the cationic adducts. The plot reported in Figure 4 describes how the percentage of the sodium adduct depends on the molecular weight (amino acids are sorted by increasing molecular weight, from left to right), dipole moment (dot sizes increase with dipole moment) [23], and binding affinity for the formation of the adduct with sodium (described by the color of the dots) [24]. Keeping in mind that potassium adducts were generally dominants, some peculiarities related to the behavior towards sodium adduct formation can be highlighted in this graph. The highest percentages of the Na adduct are referred to the amino acids having larger dipole moment and binding affinity, but the middle range of molecular weight. This could be associated with an optimal compromise between charge density and relative Na- amino acid distance. Percentage values related to sodium adduct in the range of 20–30% are common for various amino acids with different molecular weights, low to middle dipole moment and low to high binding affinity. Furthermore, it is possible to highlight some positive quasi-linear correlations considering the molecular weight, for compounds with similar dipole moment (from Ser to Cys), binding affinity (Met, His, Phe, Tyr), and both dipole moment and binding affinity (Ala, Val, Leu). Some deviations of these apparent trends, e.g., for Ile, Glu, and Trp, could be tentatively ascribed to the different polarity of the amino acidic series or to the possible steric hindrance of the side chain in the binding interaction with sodium and potassium. Excluding Met, the lowest [M+Na]^+^ relative values are associated with a combination of very low dipole moment and binding affinity, mostly irrespective of their molecular weights. An exception is surely the abnormal low Na% of Ala, which exhibited the highest [M+K]^+^ signal.

Since these observations were obtained with the background concentration of the two cations to evaluate the SWCHN surface performance e in different conditions, the concentration of sodium was increased by two orders of magnitude, by preparing the SWCHN suspension in 6 mmol/L sodium chloride water solution; under this experimental condition, it was possible to shift the ionic speciation toward the complete formation of the sodium adduct, as reported in Figure 5 for glycine, where the only detectable species was the [M+Na]^+^ ion at *m*/*z* 98.091. It is also pointed out that the increase in the salt concentration led to a significant decrease in the total ion counts due to ion suppression. A further increase of the sodium concentration to 60 mmol/L resulted in a critical ion suppression. In this case, other ionic species were detected (Figure 6) because of the formation of NaCl adducts, such as [2Na+Cl]^+^ at *m*/*z* 81.180. Under these experimental conditions, it was actually possible to note by the instrument camera the formation of “brilliant” particles on the SWCNH-based surface, putatively ascribable to NaCl crystals. These particles hindered the irradiation of the SWCNH-based surface by the laser beam and the subsequent interaction among the activated surface and analytes. In turn, these led to inefficient ionization processes on the surface, causing the observed decrease in amino acid peak intensities and the appearance of other unwanted signals.

The SWCHN-based surface was also tested in negative ionization mode. Figure 7 reports the spectra obtained for the analysis of arginine and glutamine, where the [M−H]^−^ species were clearly detectable in both cases, at *m*/*z* 173.235 and 146.123, respectively. Anyway, spectra obtained in negative mode showed a lower S/N ratio if compared to positive ionization mode, and the presence of some other unidentified ionic species.

### 2.3. Analysis of Real Oil Samples

The SWCHN-based surface was also applied to the analysis of real samples characterized by quite complex matrices. Preliminary tests were carried out on an olive oil sample for determining the fatty acid composition without pre-treating samples by the common trans-esterification protocols used in GC-FID analysis [25]. The olive oil sample was simply diluted in dichloromethane 1:1000 and spiked on the SWCNH-based surface. A typical SALDI-MS spectrum of olive oil obtained in positive ionization mode is reported in Figure 8. Besides the presence of ions at *m*/*z* 321.103 and 319.113 due to potassium adduct of oleic and linoleic acids, respectively, different triglycerides were well detectable in the 860–940 *m*/*z* range as potassium adducts. Sodium adducts were present only with low abundance, approximately <5% less than the corresponding potassium ones. It is noteworthy that the proposed SALDI approach prevents the in-source fragmentation of triglycerides to diglyceride species and the consequent presence of signals in the 500–750 *m*/*z* range, typical of MALDI and ESI (Electrospray ionization) ionization [26].

MS/MS experiments were also carried out in order to identify the fatty acid composition of the detected triglycerides. The identification of the detected species is reported in Table 1. The relative concentration of the identified fatty acids was obtained by calculating the percentage ratio between the sum of the intensities of the considered fatty acid in the different identified triglycerides and the sum of the intensities of all the fatty acids in each triglyceride. Preliminary results obtained for a commercial olive oil stated the following percentage fatty acid composition: 82.6% for oleic acids, 4.2% for linoleic acid and 12.2% for palmitic. These results are in line with previously reported data on olive oil composition [27]. Although the obtained data need to be confirmed by comparing results with the official method of analysis and extended to other types of oils, a screening of possible adulterations is certainly affordable with this approach, which is fast and cost-effective.

### 2.4. Future Perspectives

Although this work reports preliminary results, possible applications of SALDI-MS based on SWCNH surface are potentially transferable to other fields, such as environment, food and beverage, and clinic analysis. Promising applications concern the targeted analysis of amino acids or a limited number of small molecules in quite complex matrices amenable for MALDI applications, like cells, tissues, plants, etc., also avoiding time-consuming sample pre-processing. The reported application regarding the determination of the fatty acid composition of olive oil can be easily broadened to other types of oils, and also dietary supplements containing fatty acids or triglycerides, or other matrices, such as fruits or vegetables. Further applications in clinical routine may include the quantification of selected biomarkers in blood plasma, urine, saliva, etc., for the diagnosis or prognosis of diseases, or the real-time monitoring of xenobiotics in other biological specimens. The screening of priority contaminants in different environmental matrices or biomarkers or is also another potential and interesting application, as a fast and cost-effective early-warning system.

As a matter of fact, the proposed SALDI configuration may fill some gaps in the MALDI analysis of small molecules, maintaining the advantages of a quick, simple and reliable identification and semi-quantitation typical of the MALDI-MS.

## 3. Material and Methods

### 3.1. Chemicals

Analytical grade Glycine (Gly), Alanine (Ala), Valine (Val), Leucine (Leu), Isoleucine (Ile), Proline (Pro), Serine (Ser), Threonine (Thr), Asparagine (Asn), Glutamine (Gln), Cysteine (Cys), Methionine (Met), Tyrosine (Tyr), Tryptophan (Trp), Aspartate (Asp), Glutamate (Glu), Histidine (His), Lysine (Lys), Arginine (Arg), Phenylalanine (Phe) were purchased from Sigma Aldrich Italy (Milano, Italy). LC-MS grade dichloromethane was purchased from Carlo Erba Reagents (Milano, Italy). Ultrapure-grade water was produced by a Pure-Lab Option Q apparatus (Elga LabWater, HighWycombe, UK).

The single-wall carbon nano-horns (SWNHs) used in this work were produced and kindly provided by Carbonium Srl (Padova, Italy) [28]. Details on the physical characterization are reported in [16].

### 3.2. Surface Preparation and Sample Deposition

SWCNH powder was suspended in ultrapure water at the concentration of 5 mg/mL and sonicated for 1 min. Two microliters of the obtained suspension were deposited on a stainless-steel sample holder and left in the air until complete dryness. Subsequently, 1 µL of the sample was deposited on the obtained surface and dried for 20 min before the introduction into the mass spectrometer. The procedure is schematized in Appendix A.

Amino acids standards were prepared by solubilizing 1 mg of each analyte in 1 mL of ultrapure water. The olive oil sample was simply prepared by diluting 1 µL of oil (commercial olive oil produced in EU) in 1 mL of dichloromethane. Further sample pre-treatment was avoided.

### 3.3. MALDI Analysis

MALDI-MS measurements were performed using an UltrafleXtreme MALDI-TOF instrument (Bruker Daltonics, Bremen, Germany), equipped with a 1 kHz smartbeam II laser (λ = 355 nm) operating in both reflectron positive and negative ion mode.

The instrumental conditions for reflectron positive mode were ion selector 1 (IS1) = 25.00 kV, IS2 = 22.40 kV, lens = 8.00 kV, reflectron potential = 26.45 kV, and delay time = 120 ns. External mass calibration (Peptide Calibration Standard, Bruker Daltonics, Bremen, Germany) was based on monoisotopic values of [M+H]^+^ of angiotensin II, angiotensin I, substance P, bombesin, adrenocorticotropic hormone (ACTH) clip (1–17), ACTH clip (18–39), somatostatin 28 at *m*/*z* 1046.5420, 1296.6853, 1347.7361, 1619.8230, 2093.0868, 2465.1990, and 3147.4714, respectively. MS/MS experiments were performed using the LIFT device in the experimental conditions: IS1 = 7.5 kV, IS2 = 6.75 kV, LIFT 1 = 19 kV, LIFT 2 = 3.7 kV, reflector 1 = 29.5 kV, delay time = 70 ns.

The instrumental conditions for reflectron negative mode were ion selector 1 (IS1) = 20.00 kV, IS2 = 17.85 kV, lens = 6.15 kV, reflectron potential = 21.15 kV, and delay time = 80 ns. External mass calibration was based on monoisotopic values of [M−H]^−^ and their [2M−H]^−^ of 3,5-dihydroxybenzoic acid (DHB), α-cyano-4-hydroxy-cinnamic acid (CHCA) and sinapinic acid (SA) at *m*/*z* 153.0199, 188.0353, 223.0618, 307.0465, 377.077 and 447.1303, respectively.

Each sample was analyzed in triplicate. Relative intensities were calculated by the percentage ratio of the absolute intensity of the considered adduct, i.e., [M+Na]^+^ or [M+K]^+^, with respect to the sum of the absolute intensities of the [M+Na]^+^, [M+K]^+^ and [M+H]^+^ species. Results are reported as mean values and the associated relative standard deviation was always lower than 15%.

### 3.4. ICP-MS Analysis

Na and K were measured by using inductively coupled plasma coupled to a mass spectrometer (ICP-MS) Agilent Technologies 7700x ICP-MS system (Agilent Technologies International Japan, Ltd., Tokyo, Japan).

All solutions were prepared in milliQ ultrapure water obtained with a Millipore Plus System (Milan, Italy, resistivity 18.2 MΩ cm^−1^). The ICP-MS was tuned daily using a 1 μg L^−1^ tuning solution containing 140 Ce, 7 Li, 205 Tl and 89 Y (Agilent Technologies, Santa Clara, CA, USA). A 50 μg L^−1^ solution of 45 Sc and 115 In (Aristar^®^, WVR Chemicals, Leicestershire, UK) prepared in 3% (*v*/*v*) nitric acid was used as an internal standard through addition to the sample solution via a T-junction. The IS was adopted anyway for correcting both the matrix and drift effects. Multielement standard solutions for calibration were prepared in HNO_3_ 3.5% by gravimetric serial dilution at five different concentrations (min. 5 mg/L–max. 25 mg/L). All regressions were linear with a determination coefficient (R^2^) larger than 0.9998.

The samples (10 mg) were digested with 5 mL of 68% HNO_3_ and 2 mL H_2_SO_4_ 97% in a microwave system CEM EXPLORER SPD PLUS (CEM Corporation, NC, USA) at a heating rate of 30 °C/min from room temperature to 210 °C (power of 300 W). The vial pressure was 400 psi and the digestion procedure took 10 min. Samples, after the digestion step, were suitably diluted and filtered before ICP-MS analysis.

## 4. Conclusions

MALDI is a sensitive and fast MS technique for the analysis of high molecular weight compounds. Several limitations were reported for the analysis of low molecular weight analytes due to the matrix interferences. In the present work, SWCNHs were proposed as a boosting surface for different SALDI-MS applications, to overcome MALDI-MS drawbacks. The SWCNH-based surface was tested in the analysis of amino acids and triglycerides, obtaining promising, although preliminary, results. SWCNHs contained not negligible amounts of K and Na, which in turn promoted the formation of adducts instead of protonated species. The proposed surface was proved to be tolerant to salt concentration, up to 10 mmol/L, and suitable for the analysis of samples characterized by quite complex matrices. In the case of triglycerides, a limited source-fragmentation of analytes was also observed. Results indicate that SWCNH is a promising substrate for the analysis of low molecular weight compounds with different polarities by SALDI-MS approaches.

## Figures and Tables

**Figure 1 ijms-23-05027-f001:**
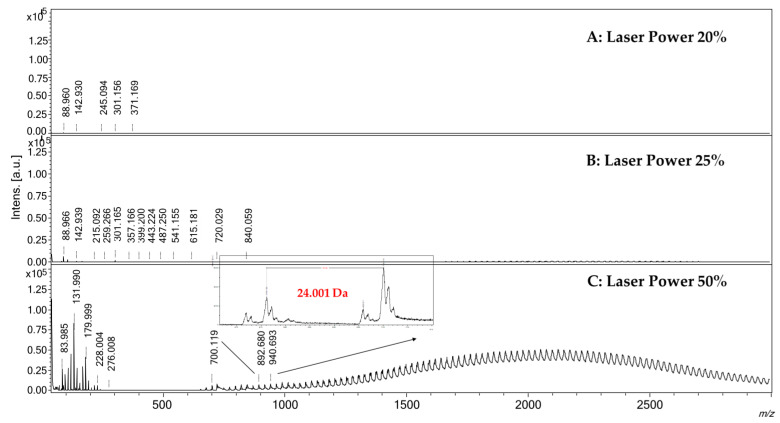
LDI-MS spectra acquired in positive ionization mode of the SWCNH-based surface using different laser power. Panel (**A**): Laser Power 20%; Panel (**B**): Laser Power 25%; Panel (**C**): Laser Power 50%. In panel (**C**) is possible to note the presence of several signals in the range 600–3000 *m*/*z*, separated by 24 mass units, due to the ionization of SWCNHs.

**Figure 2 ijms-23-05027-f002:**
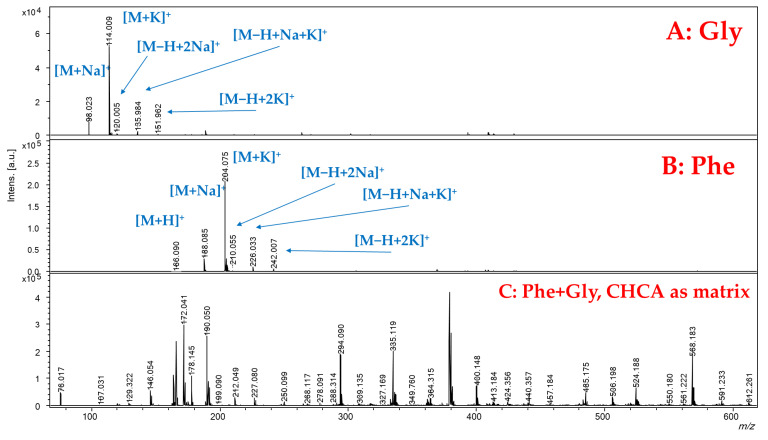
SALDI-MS spectra, obtained in positive ionization mode, for the analysis of (**A**): glycine (water solution, 1 mg/mL) and (**B**): phenylalanine (water solution, 1 mg/mL). (**C**): Spectrum in MALDI condition using CHCA as matrix, in positive ionization mode, for the analysis of Gly and Phe (water solution, 1 mg/mL each).

**Figure 3 ijms-23-05027-f003:**
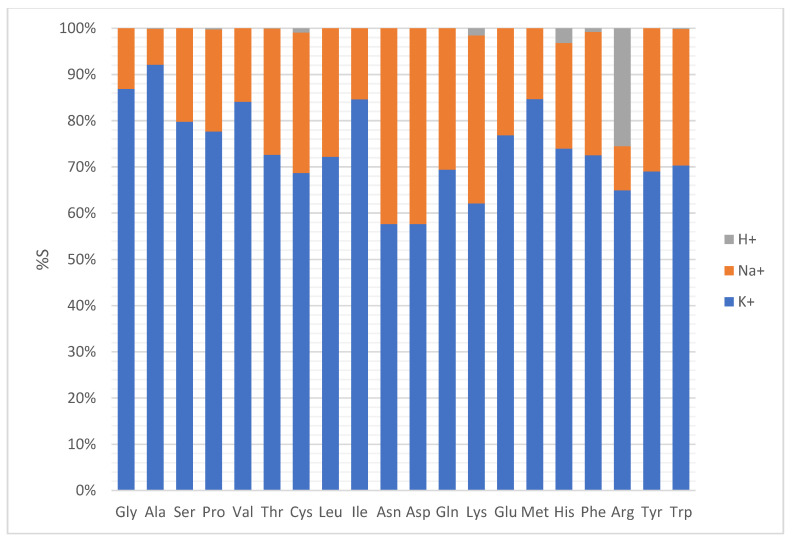
Percentage (%S) of potassium (blue), sodium (orange) and protonated (gray) adducts with respect to the sum of the three signals formed for each of the considered amino acids in SALDI condition, positive ion mode.

**Figure 4 ijms-23-05027-f004:**
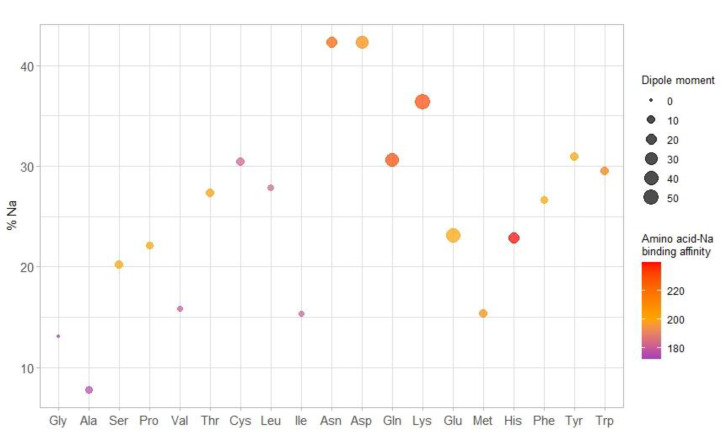
Percentage of sodium-amino acid adduct formation. Amino acids are listed by increasing molecular mass; dots are colored depending on the binding affinity between Na and the amino acid and the dimension increases with increasing dipole moment. Generally, some positive correlations between the molecular weight and the %Na were highlighted for non-polar amino acids (Gly, Ser, Pro, Thr and Cys or Ala, Val and Leu) for the same dipole moment and for Met, His, Phe and Tyr for the same binding affinity. Outliers amino acids are also present, putatively due to the arrangement of the characteristic chemical groups in the side chain.

**Figure 5 ijms-23-05027-f005:**
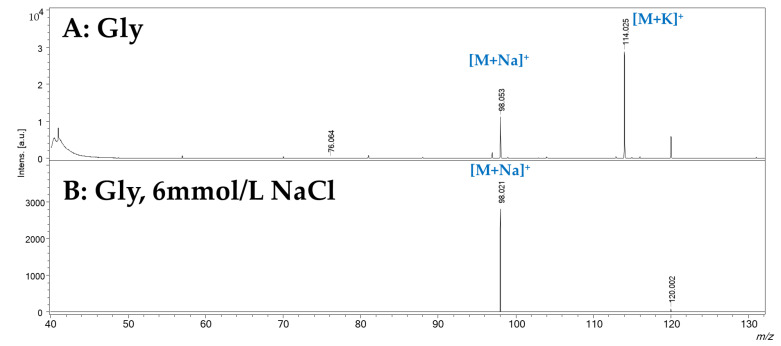
SALDI-MS spectrum, obtained in positive ionization mode by treating SWCNHs with a 6 mmol/L solution of NaCl. Panel (**A**): water solution of glycine (1 mg/mL). Panel (**B**): water solution of glycine (1 mg/mL).

**Figure 6 ijms-23-05027-f006:**
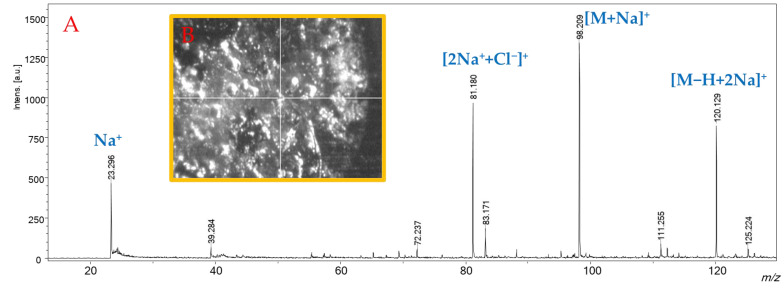
Panel (**A**): SALDI-MS spectrum, in positive ionization mode, of a water solution of glycine (1 mg/mL), obtained by treating SWCNHs with a 60 mmol/L solution of NaCl. Panel (**B**): “Brilliant” structures on the SWCNH-based surface treated with a 60 mmol/L solution of NaCl captured by the instrumental camera.

**Figure 7 ijms-23-05027-f007:**
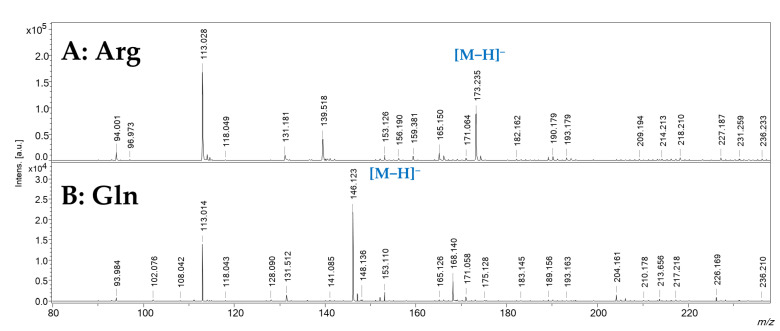
SALDI-MS spectra obtained in negative ionization mode for the analysis of panel (**A**): arginine (Arg, water solution, 1 mg/mL) and panel (**B**): glutamine (Gln, water solution, 1 mg/mL).

**Figure 8 ijms-23-05027-f008:**
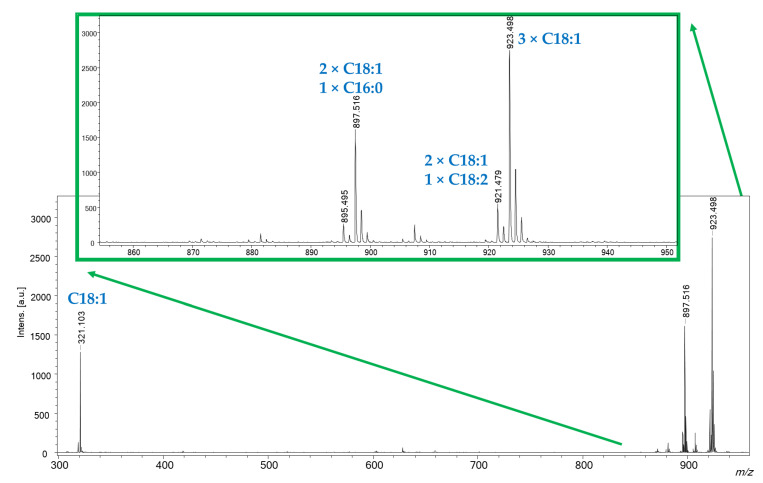
SALDI-MS spectrum of an olive oil sample, dilutes 1:1000 with dichloromethane. Identification of detected species was confirmed by MS/MS analysis.

**Table 1 ijms-23-05027-t001:** Composition in fatty acids, determined by MS/MS experiments, of the triglycerides adducts detected for the analysis of an olive oil sample by SALDI-MS.

*m*/*z*	Identified FA Composition	Adduct
923.498	3 × C18:1 (oleic acid)	[M+K]^+^
921.463	2 × C18:1 (oleic acid) 1 × C18:2 (linoleic acid)	[M+K]^+^
907.478	3 × C18:1 (oleic acid)	[M+Na]^+^
905.435	2 × C18:1 (oleic acid) 1 × C18:2 (linoleic acid)	[M+Na]^+^
897.479	2 × C18:1 (oleic acid) 1 × C16:0 (palmitic acid)	[M+K]^+^
895.457	1 × C18:1 (oleic acid) 1 × C18:2 (linoleic acid) 1 × C16:0 (palmitic acid)	[M+K]^+^
881.455	2 × C18:1 (oleic acid) 1 × C16:0 (palmitic acid)	[M+Na]^+^
879.489	1 × C18:1 (oleic acid) 1 × C18:2 (linoleic acid) 1 × C16:0 (palmitic acid)	[M+Na]^+^

## Data Availability

The data presented in this study are available on request from the corresponding author.

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
