# Peer review of "Single-Walled Carbon Nanohorns as Boosting Surface for the Analysis of Low-Molecular-Weight Compounds by SALDI-MS"

_ijms, 2022, doi:10.3390/ijms23095027_

Round 1
Reviewer 1 Report
The paper sounds well, it has merit, and is well-written.
1. Some references can be included in the manuscript:
a. Ma, Rongna, et al. "Surface‐assisted laser desorption/ionization mass spectrometric detection of biomolecules by using functional single‐walled carbon nanohorns as the matrix." Chemistry–A European Journal 19.1 (2013): 102-108.
b. Wang, J., Liu, Q., Liang, Y., & Jiang, G. (2016). Recent progress in application of carbon nanomaterials in laser desorption/ionization mass spectrometry. Analytical and bioanalytical chemistry, 408(11), 2861-2873.
c. Calandra, E., Crotti, S., Agostini, M., Nitti, D., Roverso, M., Toffoli, G., ... & Traldi, P. (2014). Matrix-assisted laser desorption/ionization, nanostructure-assisted laser desorption/ionization and carbon nanohorns in the detection of antineoplastic drugs. 1. The cases of irinotecan, sunitinib and 6-alpha-hydroxy paclitaxel. European Journal of Mass Spectrometry, 20(6), 445-459.
2. The carbon nanohorns which are used must be characterized: diameter ( TEM), surface area ( BET), RAMAN, FTIR, etc.
3. The section " Conclusions" must be detailed.
Author Response
Reviewer 1
The paper sounds well, it has merit, and is well-written.
- Some references can be included in the manuscript:
- Ma, Rongna, et al. "Surface‐assisted laser desorption/ionization mass spectrometric detection of biomolecules by using functional single‐walled carbon nanohorns as the matrix." Chemistry–A European Journal19.1 (2013): 102-108.
- Wang, J., Liu, Q., Liang, Y., & Jiang, G. (2016). Recent progress in application of carbon nanomaterials in laser desorption/ionization mass spectrometry. Analytical and bioanalytical chemistry, 408(11), 2861-2873.
- Calandra, E., Crotti, S., Agostini, M., Nitti, D., Roverso, M., Toffoli, G., ... & Traldi, P. (2014). Matrix-assisted laser desorption/ionization, nanostructure-assisted laser desorption/ionization and carbon nanohorns in the detection of antineoplastic drugs. 1. The cases of irinotecan, sunitinib and 6-alpha-hydroxy paclitaxel. European Journal of Mass Spectrometry, 20(6), 445-459.
Some references were added, as suggested by the Referee.
- The carbon nanohorns which are used must be characterized: diameter ( TEM), surface area ( BET), RAMAN, FTIR, etc.
Chemical and physical characterization of the used SWCNHs were already reported. Thus the related reference was added in the text: [16] Comisso, N.; Berlouis, L.E.A.; Morrow, J.; Pagura, C. Changes in Hydrogen Storage Properties of Carbon Nano-Horns Submitted to Thermal Oxidation. Int. J. Hydrog. Energy 2010, 35, 9070–9081, doi:10.1016/j.ijhydene.2010.06.034.
- The section " Conclusions" must be detailed.
Taking into consideration the suggestion (3) of the Referee 2 of reducing words to 150, conclusions were maintained in the original form. We hope that this compromise may be acceptable.

Reviewer 2 Report
1) Avoid using mM units. Please use mass units in order to avoid confusion. There are many such mistakes in this document. USE standard SI units.
2) State the novelty of this work in 5 sentences.
3) Write the conclusions and abstract in < 150 words, one paragraph. A lot of general text should be avoided.
4) Describe and discuss the following aspects in more detail with literature support:
- co-crystallization process of the analytes
- ionization mechanism
- analysis of polymers, antibiotics, peptides
- quantitative analysis of antineoplastic drug for pharmacokinetics purposes
5) Write details of sample pre-treatment and duplicates + STAT.
6) What are the interfering species and their roles
7) Concerning the use of common MALDI matrices, what are their limitations.
8) Discuss the role of “brilliant” particles in DETAIL.
9) Write the practical applications and future research of this work in ~ 250 words, as a separate section.
10) Conclusion = Conclusions
11) Add more REFs from 2021 and 2022. Use your keywords and search MDPI, other websites.
12)You should increase the font sizes in the X and Y axis of all the figures. They are rather small. Also, you can remove the outside box type border from some of the plots.
Author Response
Reviewer 2
- Avoid using mM units. Please use mass units in order to avoid confusion. There are many such mistakes in this document. USE standard SI units.
Done, but we prefer to add anyway the molar concentrations that can be useful for researchers working in the biochemical field, where concentrations of small molecules and macromolecules can be better evaluated and compared as molarity.
2) State the novelty of this work in 5 sentences.
Highlights:
- Single-walled carbon nanohorns as surface for the detection of low-molecular weight compounds by SALDI-MS.
- Evaluation of the experimental parameters implied in SALDI-MS performance.
- Preliminary study of the response and interferences of 20 amino acids by varying composition of the surface
- First application of SALDI-MS for the direct analysis of triglycerides in olive oil samples without derivatization.
To clarify the novelty, the following paragrapher was added at the end of the introduction: “The present work reports the optimization of a SALDI-based method for the anal-ysis of low molecular compounds with a wide range of polarity, in particular several amino acids and triglycerides. The single-walled carbon nanohorns-based surface was tested in different conditions to describe from an experimental point of view the ioni-zation processes occurring on the substrate and the combine effect of analyte and sur-face on the production of ionic adducts during the process. The method was also ap-plied to the analysis of real samples i.e. olive oil, to proposed possible future applica-tions.”
- Write the conclusions and abstract in < 150 words, one paragraph. A lot of general text should be avoided.
Abstract and conclusions were both reduced to 155 words. We hope that these modifications may be acceptable for the Editorial Office.
4) Describe and discuss the following aspects in more detail with literature support:
- co-crystallization process of the analytes
A more detailed sentence was added in the text as lines 33-36 “Furthermore, the co-crystallization process, i.e. the inclusion of analyte molecules in the matrix crystals occurring by solvent evaporation at room temperature, causes in-homogeneity of the resulted crystals, the presence of hot-spots, and the unwanted coffee-ring effect [1].”
- ionization mechanism
Modified in “The ionization mechanism implied in the production of analyte ions is still debated and not fully understood. Generally, it is reported that the laser shot promotes the formation of a laser induced plasma containing ions or activated chemical species that in turn interact with the sample, provoking the ionization of analytes. Surface plasmon effect and thermal and non-thermal processes may also be involved in the ionization process, contributing to the desorption of non-volatile analytes and to the final effi-ciency of the technique [4]”
- analysis of polymers, antibiotics, peptides
the sentence was further detailed (lines 77-79) “of polymers (polyethyleneimine and poly(ethylene glycol), pharmaceuticals (verapamil, metadone), peptides (glutathione, bradykinin 1–7 and bacitracin) and glucose…”
- quantitative analysis of antineoplastic drug for pharmacokinetics purposes
the sentence was further detailed (lines 106-109) “of antineoplastic drug (irinotecan, sunitinib, and 6-a-hydroxy-paclitaxel) for pharmacoki-netic studies in human blood plasma and personalized medicine applications …”.
5) Write details of sample pre-treatment and duplicates + STAT.
Samples were not pre-treated.
Text was integrated accordingly to this request: Each sample was analyzed in triplicate. Relative intensities were calculated by the percentage ratio of the absolute intensity of the considered adduct, i.e. [M+Na]+ or [M+K]+, with respect to the sum of the absolute intensities of the [M+Na]+, [M+K]+ and [M+H]+ species. Results are reported as mean values and the associated relative standard deviation was always lower than 15%.
6) What are the interfering species and their roles
Sentence modified in “…to depict and evaluate the presence of potential interfering signals in the low m/z range, generally due to the ionization of residual contaminants on the sample-holder or contained in solvents and chemicals used. The ionization of these species, also pos-sible in MALDI condition, may be responsible for analytes ion suppression or signal overlapping.”
7) Concerning the use of common MALDI matrices, what are their limitations.
This info was reported in the introduction; anyway, also according to the previous comment a slight modification was proposed: “The analysis of low molecular weight compounds by matrix-assisted laser desorption ionization mass spectrometry (MALDI-MS) is challenging when common organic MALDI matrices and traditional sample deposition procedures are used. The main issue is related to the presence of background interfering species, generated by the ionization of the matrix itself, which are accounted for the overlapping or suppression of analytes signals in the low mass region of the spectrum. Furthermore, the co-crystallization process of the analytes and the matrix causes the so-called coffee-ring effect, inhomogeneity of the resulted crystal and the presence of hot-spots [1]. The severity of these effects is matrix-dependent and in-terfere with the possibility of obtaining accurate quantitative results and good reproduci-bility by MALDI-MS analysis”
8) Discuss the role of “brilliant” particles in DETAIL.
These particles hindered the irradiation of the SWCNH-based surface by the laser bean and the subsequent interaction among the activated surface and analytes. In turn, these led to inefficient ionization processes on the surface, causing the observed decrease of amino acid peak intensities and the appearance of other unwanted signals.
9) Write the practical applications and future research of this work in ~ 250 words, as a separate section.
A section titled Future perspectives was added to the paper (ca 200 words)
“Although this work reports preliminary results, possible applications of SALDI-MS based on SWCNH surface are potentially transferable to other fields, such as environment, food & beverage, and clinic analysis. Promising applications concern the targeted analysis of amino acids or a limited number of small molecules in quite complex matrices amena-ble for MALDI applications, like cells, tissues, plants etc, also avoiding time-consuming sample pre-processing. The reported application regarding the determination of the fatty acid composition of olive oil can be easily broaden to other types of oils, and also dietary supplements containing fatty acids or triglycerides, or other matrices, such as fruits or vegetable. Further applications in clinical routine may include the quantification of se-lected biomarkers in blood plasma, urine, saliva, etc., for the diagnosis or prognosis of diseases, or the real-time monitoring of xenobiotics in other biological specimens. The screening of priority contaminants in different environmental matrices or biomarkers or is also another potential and interesting application, as a fast and cost-effective ear-ly-warning systems.
As a matter of fact, the proposed SALDI configuration may fill some gaps of the MALDI analy-sis of small molecules, maintaining the advantages of a quick, simple and reliable identi-fication and semi-quantitation typical of the MALDI-MS.”
10) Conclusion = Conclusions
Done
11) Add more REFs from 2021 and 2022. Use your keywords and search MDPI, other websites.
Done
12)You should increase the font sizes in the X and Y axis of all the figures. They are rather small. Also, you can remove the outside box type border from some of the plots.
Done
